# ON ROBUSTNESS-ACCURACY CHARACTERIZATION OF LARGE LANGUAGE MODELS USING SYNTHETIC DATASETS

## ABSTRACT

In recent years, large language models (LLMs) that were pretrained at scale on diverse data have proven to be a successful approach for solving different downstream tasks. However, new concerns about proper performance evaluation have been raised, especially for test-data leakage caused by accidentally including them during pretraining, or by indirectly exposing them through API calls for evaluation. Motivated by these, in this paper, we propose a new evaluation workflow that generates steerable synthetic language datasets and proxy tasks for benchmarking the performance of pertrained LLMs on sentence classification tasks. This approach allows for better characterization of the joint analysis on the robustness and accuracy of LLMs without risking sensitive information leakage. It also provides a more controlled and private way to evaluate LLMs that avoids overfitting specific test sets. Verified on various pretrained LLMs, the proposed approach demonstrates promising high correlation with real downstream performance.

## 1 INTRODUCTION

In recent years, large language models (LLMs) have emerged, showcasing remarkable capabilities across a wide range of natural language processing (NLP) applications (Peters et al., 2018; Devlin et al., 2019; Yang et al., 2019; Raffel et al., 2020; Rae et al., 2021; Wei et al., 2022a; Thoppilan et al., 2022; Hoffmann et al., 2022; Chowdhery et al., 2022). To empower LLMs with more generic abilities (zero-shot, reasoning, and generalization), notable progress has been made in proposing prefix-tuning (Li & Liang, 2021), instruction-tuning (Wei et al., 2022a), scratchpad prompting (Nye et al., 2021; Lewkowycz et al., 2022; Wei et al., 2022c), etc. As we continue to scale up LLMs to critical sizes, they often exhibit emergent abilities (Wei et al., 2022b) that they are not directly trained to have, including performing arithmetic, answering questions, summarizing passages, and more.

While new opportunities present themselves with foundation models, they also bring forth potential risks and challenges (Bommasani et al., 2021; Blodgett & Madaio, 2021; Wiggins & Tejani, 2022; Thieme et al., 2023; Biderman et al., 2023). For example, despite the unprecedented publicity of LLMs and beliefs in their emergent abilities, some also argued the emergent abilities of LLMs are a mirage (Schaeffer et al., 2023) and a change in metric choice can lead to a different conclusion. Recently, researchers have also openly expressed concerns about the potential for language models to be trained on test sets (Liang, 2023). Even worse, private or held-out unpublished test sets may as well be vulnerable to data leakage through querying the LLMs via APIs for evaluation purposes. Extraction attacks (Carlini et al., 2019; 2021), membership inference attacks (Hisamoto et al., 2020; Thomas et al., 2020; Mireshghallah et al., 2022), and generative embedding inversion attack (Li et al., 2023), caused by unintended memorization (Carlini et al., 2019; Biderman et al., 2023) further deepened our concerns about privacy leakage during test time.

To address this caveat of "information leakage" during test time, in this paper, we aim at proposing a new testbed for benchmarking LLMs with synthetic data. We design this testbed to serve as a "minimum" model test for two basic skills infants must learn when acquiring language, identifying (sentiments in) words and linguistic structures (Frost et al., 2020). To achieve these, we (1) create a basic artificial language with some long-range dependency structure to mimic statistical learning of language, (2) create class labels associated with different notions (sentiment or some other attributes),

Figure 1: Overview of SynTextBench. SynTextBench generates a set of synthetic datasets from any given lexicon with word-level labels. We test the given LLM on sentence-level tasks with these datasets and obtain robustness-accuracy characterization under a range of steerable task difficulties. For each LLM, we can plot the robustness-accuracy trade-off curve and make model comparisons.

and then (3) probe LLMs on that. Specifically, we leverage existing sentiment lexicons, such as SentiWordNet 3.0 (Baccianella et al., 2010), to generate working word lists based on the word (or synset) level labels. We build positive, negative, and neutral word lists from SentiWordNet 3.0, and constructing sentences following the nesting parentheses (Papadimitriou & Jurafsky, 2020), which mimics the recursion structural hypothesis about the narrow language faculty in humans (Hauser et al., 2002) and the dependency tree structure in natural language (Chiang & Lee, 2022). By maneuvering the mixing percentage of binary words (positive/negative words) and neutral words, we create a configurable testbed for evaluating the performance of LLMs on different levels of difficulty and complexity. Finally, we benchmark and quantify the ability of each LLM on sentence classification tasks by comparing their performance on a set of our synthetic datasets with varying difficulty levels.

We dub our evaluation framework using synthetic data by *SynTextBench* and present the workflow in Figure 1, where we focus on benchmarking LLM sentence embeddings in terms of their accuracy and robustness. By accuracy, we are interested in analyzing the linear separability of sentence representations rendered by different pretrained LLMs. We note that in learning sentence embeddings, the go-to metrics are cosine distance or linear probing accuracy, both of which imply separability. By robustness, we refer to the decision margin on these sentence embeddings with respect to the optimal classification strategy. We derive both measures using only the constructed synthetic datasets, which allow for privacy-preserving benchmarking of LLMs. SynTextBench is designed as an extendable framework for the evaluation of language sentence representations that covers a range of controllable task difficulties. We envision our framework to also be easily adaptable to other societal properties such as value assessment and will facilitate independent and sustainable LLM auditing (Weidinger et al., 2021; Ganguli et al., 2022; Madiega, 2021; Mökander et al., 2023; Rastogi et al., 2023).

Our **main contributions** are:

• We introduce *SynTextBench*, a novel theoretically-grounded framework to generate steerable synthetic datasets towards a holistic evaluation of LLMs. The use of synthetic datasets alleviates the risk of test-data leakage and offers new tools for LLM testing and auditing.

• SynTextBench provides a configurable lightweight testbed and a quantifiable metric for evaluating the robustness and accuracy of LLMs on different levels of difficulty and complexity for sentence classification tasks, with no restrictions on the model architecture.

• We conduct experiments with several state-of-the-art LLMs on our testbed and report their performance and behavior. SynTextBench, as a real-data-free evaluation method, shows high correlation with robustness-accuracy performance evaluated on real data. Further study demonstrates its capability of making quick attribution comparisons such as analyzing fine-tuning effects for LLMs.

## 2 METHODOLOGY

### 2.1 WHY USING SYNTHETIC DATASETS FOR LLM EVALUATION?

To reduce the reliance on real-world data, we propose to build synthetic NLP tasks by generating synthetic sentences as model inputs at test time. This way, we no longer need to exchange sensitive private data or label-annotated data as test sets with LLM APIs. In making a steerable and transparent evaluation framework for LLMs, we first detail the desiderata of proxy tasks and the evaluation metric.

• Task substance: Tasks should test a pretrained LLM's ability to encode sentence representations that preserve class separability when evaluated by a linear classifier.

• Task difficulty: Tasks' difficulty should be configurable to allow for comprehensive analysis, i.e., one can generate tasks of various levels of difficulty.

• Task feasibility: Tasks should be feasible to solve, i.e., the sentences should be distinguishable to

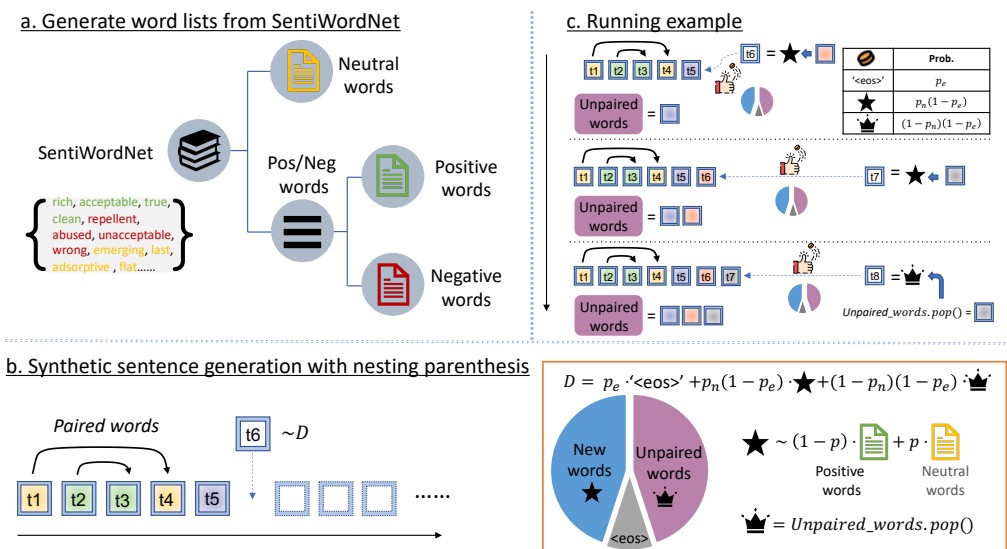

Figure 2: Overview of the sentence generation procedure. In block a, we generate word lists from SentiWordNet 3.0. In block b, we generate each sentence token following nesting parenthesis and mixing distribution $D$. In block c, we show a running example of sequentially generating $t_6, t_7, t_8$.

a certain degree by an algorithm that works on the raw sentences input.

- Task independence: Tasks' ground-truth should be independent of the LLM to be evaluated, in order to avoid biased evaluation, e.g., the label in the task should not be given by an LLM.
- Task equity: Tasks should be able to be generated by anyone and affordable for anyone without requiring any private data or favoring any party with more resources.
- Metric informativeness: The designed framework should give a quantifiable metric that has a clear implication (e.g., the larger the better) and correlates well with the real performance.

With these in mind, it is straightforward to see why we should not opt for synthetic datasets generated by any LLM: (1) task difficulty would not be configurable, (2) the evaluation might favor the LLM that generates the synthetic sentences and/or the pseudo-labels (causing label leakage), and (3) any auditor without access to proprietary LLMs or datasets cannot run independent evaluation.

In the following, we explain how we leverage sentiment lexicons, such as SentiWordNet 3.0, to create building blocks for our framework. Then, we put together building blocks and generate synthetic inputs to LLMs by observing a nesting structure. We adjust the mixing ratio of ingredients in the recipe to simulate tasks of different difficulties. We depict this procedure in Figure 2. Finally, we will introduce our evaluation workflow and how we arrive at a quantifiable metric.

## 2.2 CONSTRUCTING SYNTHETIC DATASETS AND TASKS

**Word List.** Building a synthetic task requires us to define the synthetic inputs to be used. Here, we utilize sentiment lexicons with word-level labeling. SentiWordNet labels the synsets of WORD-NET (Miller, 1995) according to the notions of "positivity", "negativity", and "neutrality". Each of the entries in SentiWordNet has PosScore and NegScore denoting the positivity and negativity score, and ObjScore is calculated by 1 - (PosScore + NegScore), denoting the neutrality score. When categorizing these words, we remove the sense number associated with the words and group words into individual word list based on the following criteria: for a word $w$,

- if PosScore > NegScore, we categorize $w$ into the positive word list;
- if PosScore < NegScore, we categorize $w$ into the negative word list;
- if PosScore = NegScore = 0, we categorize $w$ into the neutral word list.

We give running examples in the Appendix Table 5 for better understanding. In practice, we perform the procedure on SentiWordNet 3.0 and gather a positive word list with 23147 words, a negative word list with 26440 words, and a neutral word list with 154993 words. The same procedures can be applied to any sentiment lexicons with word-level labeling, which will result in different word lists. To this end, we created the word lists from SentiWordNet 3.0 as depicted in Figure 2(a).

**Sentence structure.** A recent literature (Papadimitriou & Jurafsky, 2020) explored the power of music and Java code in training models that transfer to NLP tasks. It further stated that, not only music and Jave code, non-linguistic artificial parentheses languages can also train language models

that yield substantial gains compared to random data when testing on natural language (Chiang & Lee, 2020; Ri & Tsuruoka, 2022; Papadimitriou & Jurafsky, 2023). Motivated by this, we follow one of the abstract structures, nesting parenthesis, when generating the synthetic sentences in our proxy tasks. The inclusion of the parenthesis is to guarantee we test for the linguistic structures, whose importance is repeatedly advocated in literature from both machine learning and cognitive science (Frost et al., 2020; Wilson et al., 2020; Manning et al., 2020). Specifically, nesting parenthesis involves paired tokens and a recursive structure. For example, by referring to Figure 2(b), one sees that $t_1$ and $t_4$ are paired words, while $t_2$ and $t_3$ are another paired words. In our example, the words are hierarchically nested, meaning the token to be paired with $t_2$, which is $t_3$ in our case, should appear before the pairing token with $t_1$. In other words, it observes a "last in first out" data structure, and the arcs in Figure 2(b) do not cross.

**Sentence generation and difficulty level.** With the created word list from above, we will now explain how to do sentence generation following the structure introduced. Let us revisit the case in Figure 2(b). Assume we want to generate a positive sentence (label $y = 1$), and we already generated the first five tokens $t_1 : t_5$ in the sentence with colors denoting the picked word. Now, to decide the next token, we sample $t_6$ from a mixing distribution $D$, where

$$D = p_e \cdot \text{`<eos>'} + p_n(1 - p_e) \cdot \text{last\_unpaired\_word} + (1 - p_n)(1 - p_e) \cdot D_{\text{new}}. \qquad (1)$$

To interpret distribution $D$, we realize that there are essentially 3 possible outcomes for the incoming $t_6$ token: (1) it can be the end of sentence indicator '<eos>', (2) it can be the popped token from the stack that stores the unpaired words, i.e., the last unpaired word, (3) it can be a new word. If it is to pick a new word, this word will be sampled from the distribution of new words $D_{\text{new}}$, which directly depends on the label $y$ of the sentence to be generated and the desired task difficulty. For a positive sentence ($y = 1$), $D_{\text{new}|y=1}$ is described by the probability density function (PDF) $p \cdot f_{\text{NEU}}(x) + (1 - p) \cdot f_{\text{POS}}(x)$, where $p$ specifies the percentage of neutral words in a synthetic sentence, $f_{\text{NEU}}$ gives the PDF of neutral words, and $f_{\text{POS}}$ gives the PDF of positive words. Similarly, if we are to generate a negative sentence ($y = -1$), we have $D_{\text{new}|y=-1}$ described by $p \cdot f_{\text{NEU}}(x) + (1 - p) \cdot f_{\text{NEG}}(x)$, where $f_{\text{NEG}}$ gives the PDF of negative words.

In Figure 2(c), we show a running example of the sentence generation process, where we flip a coin with 3 outcomes each time to decide on a new token. When the realization is "new words" (like in $t_6$ and $t_7$), this word will also be pushed to the stack "*Unpaired\_words*" that stores unpaired words. When we are deciding $t_8$, we draw "unpaired words" and hence $t_8$ is determined by *Unpaired\_words.pop()*. In essence, with the generated sentence, its label is determined by construction, which guarantees the **task independence** since the label is not given by an LLM. It also allows configurable **task difficulty** by adjusting the percentage $p$ of neutral words in a synthetic sentence. That is, it is easier to predict the sentiment of sentences consisting of 90% positive words and 10% neutral words than that of sentences constructed all by neutral words. On the whole, by fixing a mixing ratio $p$, together with the fixed $p_e$ and $p_n$ given in the above, one synthetic dataset will be constructed as well as a resulting proxy sentiment classification task. By varying the mixing ratio $p$, a set of tasks with diverse difficulties can be created. In the Appendix Figure 5, we prove the **task feasibility** by demonstrating the separability of generated synthetic datasets by SentiWordNet sentiment analysis algorithm (Denecke, 2008). With an increasing mixing ratio $p$, while the task becomes harder, we show there at least exists an algorithm that can separate the data to a certain degree, showcasing a lower bound on the optimal classification strategy. By our workflow of constructing synthetic datasets and tasks, we also guarantee **task equity** since the generation process requires no access to any LLM or private data, and can be readily replicated by anyone with limited resources. Furthermore, we note that the construction of synthetic datasets and tasks described herein is also extendable to other lexicons and tasks by swapping the lexicon used for extracting word lists.

Lastly, we note that during the construction of synthetic sentences, the probability $p_e$ associated with the special token '<eos>' is determined by its frequency in the English Wikipedia corpus. For the remaining mass $1 - p_e$, $p_n$ portion is assigned to new words, with its value picked following Papadimitriou & Jurafsky (2020), which is $p_n = 0.5$. Additionally, when there are no unpaired words in the stack (e.g., when drawing the starting token of the sentence, or when all the unpaired words are popped), we assign its probability $p_n(1 - p_e)$ to new words. We show the length profile of our synthetic data in Figure 4 in Appendix.

**Discussions.** The inclusion of parenthesis in our sentence structure guarantees we test for the linguistic structures but at the same time makes non-grammatical test sets. While grammar might be

---

**Algorithm 1** Benchmarking LLMs using synthetic datasets (*SynTextBench*)

---

**Input**: Sentiment lexicons $S$, a range of difficulty levels $P$, an LLM $g$, threshold accuracy $a_T$.
**Output**: SynTextBench score that quantifies the robustness-accuracy performance.

1:  Construct positive/negative/neutral word lists from sentiment lexicon $S$.
2:  **for** $p$ in $P$ **do**
3:      Generate a synthetic binary classification task and obtain training set $(x^{train}, y^{train})$ and test set $(x^{test}, y^{test})$.
4:      Calculate transformation $T_1$ and $T_{-1}$ from $z_1^{train} = \{g(x) \mid (x,y) \in (x^{train}, y^{train}), y = 1\}$ and $z_{-1}^{train} = \{g(x) \mid (x,y) \in (x^{train}, y^{train}), y = -1\}$.
5:      Transform training set and test set $\hat{z_1}^{train} = T_1(z_1^{train})$, $\hat{z_{-1}}^{train} = T_{-1}(z_{-1}^{train})$ and $\hat{z_1}^{test} = T_1(z_1^{test})$, $\hat{z_{-1}}^{test} = T_{-1}(z_{-1}^{test})$.
6:      Derive the Bayes optimal classifier $f$ according to $\text{sign}(\tilde{\mu}^T(\hat{z} - \frac{\mu_1 + \mu_2}{2}))$ based on $\hat{z_1}^{train}$ and $\hat{z_{-1}}^{train}$, i.e. $\mu_1 = \text{mean}(\hat{z_1}^{train})$, $\mu_2 = \text{mean}(\hat{z_{-1}}^{train})$.
7:      Read out the accuracy $a$ of $f$ on $\hat{z_1}^{test}$ and $\hat{z_{-1}}^{test}$, and calculate the average scale margin $\delta := avg(\|\bar{\Delta}_z\|_2)$ according to $\|\bar{\Delta}_z\|_2 = \frac{|(\hat{z} - \frac{\mu_1 + \mu_2}{2})^T \tilde{\mu}|}{\|\tilde{\mu}\|_2^2}$ for correctly-classified sentence embeddings.
8:      Denote the accuracy and average margin pair on the task by $(a_p, \delta_p)$.
9:  **end for**
10: Define a goodness function $s(a) = \frac{1}{|P|} \sum_{\{p \in P, a_p > a\}} \delta_p$, for $a \in \mathbb{R}[0, 1]$.
11: SynTextBench score $= \int_{a_T}^{1} s(a) da$.

---

crucial in some NLP tasks that requires more advanced reasoning. For sentiment analysis, we believe it should not have a strong dependency on grammar (we exclude the scenario of negation which can be detected by a rule-based method). For example, the reviews "love love fantastic", "love fantastic love" and their word permutations should all be predicted as positive, regardless of their grammar. We support this intuition by additional experiment where we noticed that 86% of the labels given by Huggingface sentiment analysis pipeline on product reviews classification (Hu & Liu, 2004) remain the same after removing 284 stop words (listed in the Appendix A.2) from the sentences and hence making them non-grammatical. We leave more details and sentence examples to Appendix A.4.

## 2.3 ROBUSTNESS-ACCURACY EVALUATION

Given an LLM $g$, let $x, y$ be the input sentence and its label, $z$ be the sentence embeddings $z = g(x) \in \mathbb{R}^n$, we are interested in evaluating the accuracy of the sentence embedding classifiers $f$, and the average distance $\Delta$ from sentence embeddings to the linear classifiers (i.e., decision margins). We let $z_1$ be $\{z : z = g(x), y = 1\}$ and $z_{-1}$ be $\{z : z = g(x), y = -1\}$.

**Preparing sentence embeddings.** Recall that Bert-flow (Li et al., 2020) and Bert-whitening (Su et al., 2021) transformed the sentence embeddings into an isotropic Gaussian distribution to remedy the anisotropic behavior in the sentence embedding vector space. We thereby also perform whitening on sentence representations before we draw the decision rule on the embeddings. Transforming a set of sentence embeddings of a class into an isotropic Gaussian involves two steps: (1) model the mean $b_y$ and covariance $\Sigma_y$ of original embeddings $z_y$, (2) apply a transformation to the embeddings $F^T S^{-1/2} z_y$, where $FSF^T = \Sigma_y$ is the singular value decomposition of $\Sigma_y$. Nevertheless, since $\Sigma_y$ can be ill-conditioned, directly applying $S^{-1/2}$ on embeddings $z_y$ might amplify noisy signals due to numerical instability. Thus, we propose to reduce the dimension according to energy-preservation (Leskovec et al., 2020) (also called variance-based methods by Falini (2022)). We select to keep $K$ dimensions according to $\arg\min_k \frac{\sum_{i=1}^{k} s_i}{\sum_{i=1}^{n} s_i} \geq 0.99$, where $s_i = \text{diag}(S)[i]$ is the $i$-th largest singular value of $S$. Till now, we see that the sentence embeddings are transformed to an $\mathbb{R}^K$ vector space via $F_{:,1:k}^T S_{1:k,1:k}^{-1/2} z_y$. We perform these operations for both classes ($y = 1$ and $y = -1$) separately. Since we want the transformed embeddings to observe the original relative distance between two classes, we further scale the distance between two whitened Gaussians by $d_{\text{Inter-class}} / d_{\text{Intra-class}}$, where the numerator $d_{\text{Inter-class}} = \|b_1 - b_{-1}\|$ calculates the inter-class distance (the distance between two class centers $b_1$ and $b_{-1}$), and the denominator

$d_{\underline{\text{Intra-class}}} = \frac{1}{m_1+m_2}(\sum_{i=1}^{m_1}\|z_1^i - b_1\| + \sum_{j=1}^{m_2}\|z_{-1}^j - b_{-1}\|)$ calculates the intra-class distance (the average distance from class data to class mean) with $m_1$ and $m_2$ being the number of positive sentences and negative sentences, respectively. We let $T_y$ denote the overall transformation operations and obtain transformed embeddings $\hat{z_1} = T_1(z_1)$ and $\hat{z_{-1}} = T_{-1}(z_{-1})$.

**Decision margins induced by robust Bayes optimal classifiers.** Recall that robust Bayes optimal classifiers explicitly give the optimal classification strategy for class-conditional Gaussian distribution in the presence of data perturbations (Bhagoji et al., 2019; Dan et al., 2020). Here, we see that $(\hat{z}, y)$ are modeled as $P_{\mu_1,\mu_2,I_K}$: $\hat{z}|y = 1 \sim \mathcal{N}(\mu_1, I_K), \hat{z}|y = -1 \sim \mathcal{N}(\mu_2, I_K)$, and $y \in \mathcal{C} = \{+1, -1\}$. While finding the robust Bayes optimal classifier generally involves solving the optimization problem $\arg\min_{\|z\|_2\leq\epsilon}(\mu - z)^T\Sigma^{-1}(\mu - z)$ (cf. Appendix A.1), we can prove that, when the covariance is an identity matrix, the class priors $\mathbb{P}(y = 1) = \tau, \mathbb{P}(y = -1) = 1-\tau$, the perturbation radius $\epsilon$, then the optimal classifier is given as simply $f : \text{sign}(w^T(\hat{z} - \frac{\mu_1+\mu_2}{2}) - q/2)$, where $q = \log\{(1 - \tau)/\tau\}$, $w = \tilde{\mu}(1-\epsilon/\|\tilde{\mu}\|_2)$, and $\tilde{\mu} = \frac{\mu_1-\mu_2}{2}$. Furthermore, when the classes are balanced (i.e., $\tau = 1/2$), the robust Bayes optimal classifier overlaps with the Bayes optimal classifier. That is, the (robust) Bayes optimal classifier is plainly $\text{sign}(\tilde{\mu}^T(\hat{z} - \frac{\mu_1+\mu_2}{2}))$, which is independent of $\epsilon$. We then use this given classifier to calculate the accuracy on the synthetic datasets. In fact, we prove in Appendix A.7 that, as long as $\tilde{\mu}$ lies completely within a degenerate subspace of the eigenspace of the covariance matrix (i.e., with eigenpairs $\{(\lambda_k, v_k), k \in [n]\}$, for $\forall i, j \in \{k : \lambda_k \neq 0, \tilde{\mu}^T v_k \neq 0\}, \lambda_i = \lambda_j = \lambda$), the $\epsilon$-robust Bayes optimal classifiers overlap for all $\epsilon$. In the case of an identity covariance matrix, the degenerated subspace of the eigenspace expands the whole $\mathbb{R}^K$, hence $\tilde{\mu}$ lies in the space naturally.

Now that we have specified the optimal robust classification rule on the transformed sentence embeddings, we write out the decision margin induced by the classifiers using an informal but more intuitive statement: For any sample $z$, the Bayes optimal classifier $f$ of class-balanced class-conditional Gaussian distribution $P_{\mu_1,\mu_2,I_K}$, yields a decision margin of $\|\Delta\|_2 = \frac{|(\hat{z}-\frac{\mu_1+\mu_2}{2})^T\tilde{\mu}|}{\|\tilde{\mu}\|_2}$, and if we scale the margin by the distance between two Gaussian centers, we obtain a scaled margin of $\|\bar{\Delta}_z\|_2 = \frac{|(\hat{z}-\frac{\mu_1+\mu_2}{2})^T\tilde{\mu}|}{\|\tilde{\mu}\|_2^2}$. We give the formal results for the generic class prior in Appendix A.7. To this end, we have prepared sentence embeddings and specified the way of calculating decision margins induced by a robust Bayes optimal classifier. In the following, we will state the complete algorithm for characterizing robustness-accuracy performance of LLMs using synthetic datasets.

## 2.4 SYNTEXTBENCH SCORE AND ALGORITHM

With Section 2.2 and Section 2.3, we now can simulate synthetic tasks of a configured level of difficulty and evaluate their accuracy and margin. In our benchmarking process, we essentially build on this foundation to generate a sequence of tasks with different difficulty levels and inspect how the magnitude of decision margins changes with the classifier accuracy. In terms of robustness-accuracy characterization, it is desirable for an LLM to consistently yield high classification accuracy, while maintaining a big decision margin (that is, less sensitive to perturbations in the embedding space). The pseudocode of the proposed framework, *SynTextBench*, is given in Algorithm 1.

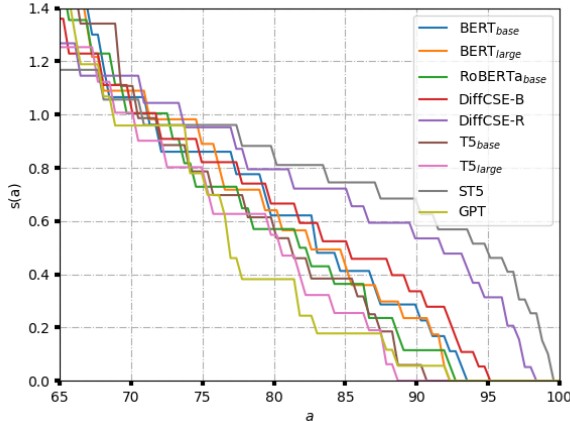

Figure 3: The goodness function $s(a)$ of nine pretrained language models. The SynTextBench score is calculated by the area under the curve.

In practice, we let $P = \{0, 0.05, \ldots, 0.9, 0.95\}$, and subsequently generate 20 synthetic datasets with $p = 0$ being the easiest and $p = 0.95$ being the hardest (cf. Section 2.2). Then, we perform analysis on the sentence embeddings of various synthetic datasets, and threshold the accuracy at $a_T$ based on utility. The threshold serves as a penalty for poor sentence embeddings that lead to an undesirable accuracy under this threshold, matching our **task substance** of testing LLM's ability to preserve

linear separability. By referring to Figure 1, Line 1 in Algorithm 1 determines the word lists from a given lexicon. From Line 2 to Line 9, the for-loop generates one synthetic dataset at one time, on which we compute an (accuracy, average margin) pair $(a_p, \delta_p)$ and draw one point on the margin-accuracy 2D plot as in Figure 1. We apply Algorithm 1 on various models and obtain a margin-accuracy curve for each model. Since we not only care about the curvature of the curve but also how the (accuracy, average margin) pairs span on the curve, we define a goodness function $s(a) = \frac{1}{|P|} \sum_{\{p \in P, a_p > a\}} \delta_p$ on $\mathbb{R}[0, 1]$ in Line 10 to account for the span. By our definition, $s(a)$ will be a monotonically decreasing function (e.g., Figure 3) and calculate the expected margin conditioned on the accuracy level. The final SynTextBench score is defined by the integration over the desirable range of threshold accuracy in Line 11, i.e. SynTextBench score $= \int_{a_T}^{1} s(a)da$. We use SynTextBench as a quantifiable score to inform the accuracy-robustness aspect of a pretrained LLM. In the later section, we will demonstrate the **metric informativeness** by measuring the correlation between SynTextBench scores and the average real-world sentence classification task performance.

## 3 EXPERIMENTS

### 3.1 SETUPS

**LMs.** In the following, we list a few pretrained LLMs predominantly considered by the sentence embedding literature Gao et al. (2021); Su et al. (2021); Chuang et al. (2022). We will showcase the use of SynTextBench on these models before moving to larger models such as LLaMA and OPT.
- BERT$_\text{base}$ and BERT$_\text{large}$ (Bidirectional Encoder Representations from Transformers (Devlin et al., 2019)) are encoder-only transformers pretrained with masked language model and next sentence prediction pre-training objectives.
- RoBERTa$_\text{base}$ (Robustly Optimized BERT Pretraining Approach (Liu et al., 2019)) is a modification of BERT that trained with dynamic masking, large mini-batches, a larger byte-level byte pair encoding, and removed the next sentence prediction objective.
- DiffCSE-B and DiffCSE-R (Difference-based Contrastive Learning for Sentence Embeddings (Chuang et al., 2022)) are BERT$_\text{base}$ and RoBERTa$_\text{base}$ models that further trained with difference-based contrastive learning.
- T5$_\text{base}$ and T5$_\text{large}$ (Text-to-Text Transfer Transformer (Raffel et al., 2020)) are encoder-decoder transformers that cast all NLP tasks into a text-to-text problem.
- ST5 (Scalable sentence encoders from pre-trained text-to-text models (Ni et al., 2022)) is initialized by T5$_\text{base}$ and trained by two-stage contrastive learning.
- (GPT) DialogRPT (Dialog Ranking Pretrained Transformers (Gao et al., 2020)) is a decoder-only model trained on vast human feedback data.

For models that have an encoder component (encoder-only or encoder-decoder), we use the average output from the first and the last layer as sentence embeddings. For the decoder-only model, we use the embedding of the last token as sentence embeddings.

**Baselines.** We followed the open-source implementation of the literature (Whitney et al., 2020) and fed the pretrained LLMs with synthetic texts generated according to Section 2.2 and reported the validation accuracy (Val loss), minimum description length (MDL), surplus description length (SDL), and $\epsilon$-sample complexity ($\epsilon$SC) as baselines (Blier & Ollivier, 2018; Voita & Titov, 2020; Whitney et al., 2020). Since these methods take one dataset as inputs, we choose a relatively easy synthetic proxy task generated by $p = 0.2$ as the input dataset.

**Objectives.** Through the experiments, our main aim is to verify the feasibility of making performance assessments of possible downstream tasks by real-data-free evaluation methods. To achieve this, we will compare the Pearson correlation coefficients of assessments given by different real-data-free evaluation methods with the performance on real-world tasks. Since SynTextBench is intended to inform the robustness-accuracy performance, we will report both the accuracy and robustness on real-world tasks for studying correlation. We use PWWS attack (Ren et al., 2019) through TextAttack, a Python framework for adversarial attacks in NLP, to generate attacks. Essentially, the attacker will perturb the inputs gradually by changing more and more words until the perturbation leads to a wrong classification result. Therefore, we report the average percentage of perturbed words in a successful attack as an indicator of the level of model robustness. As more attentions have been drawn to in-context learning (ICL) setups lately, we will also give an example where we extend SynTextBench to ICL and perform ICL on our synthetic tasks using LLMs. Finally, we will also

Table 1: Correlation between real-data-free evaluation metric and real-data accuracy at different synthetic dataset sizes.

| n | 4096 | 8192 | 16384 | 32768 |
|---|---|---|---|---|
| Val loss | 0.29±0.50 | 0.65±0.00 | 0.61±0.01 | 0.27±0.02 |
| MDL | 0.57±0.11 | 0.52±0.04 | 0.51±0.03 | 0.48±0.03 |
| SDL, $\varepsilon$=1 | 0.57±0.11 | 0.51±0.04 | 0.43±0.02 | 0.31±0.01 |
| $\varepsilon$SC, $\varepsilon$=1 | - | - | - | -0.04±0.000 |
| SynTextBench | **0.94±0.01** | **0.97±0.01** | **0.96±0.00** | **0.93±0.00** |

Table 2: Aggregated correlation with real-data-free evaluation metrics and the aggregated robustness-accuracy performance, and its breakdown.

| Correlation. w/ | Rob.-Acc. | Rob.-STS | Rob.-Transfer |
|---|---|---|---|
| Val loss | -0.06±0.15 | 0.08±0.13 | -0.13±0.24 |
| MDL | 0.64±0.06 | 0.55±0.08 | 0.62±0.03 |
| SDL, $\varepsilon$=1 | 0.60±0.02 | 0.51±0.04 | 0.58±0.028 |
| $\varepsilon$SC, $\varepsilon$=1 | - | - | - |
| SynTextBench | **0.76±0.04** | **0.76±0.03** | **0.69±0.05** |

demonstrate how SynTextBench can be used to do attribute comparisons. We defer experimental details to the appendix due to the page limit.

## 3.2 PERFORMANCE EVALUATION AND DISCUSSION

We evaluate models listed in Section 3.1 by SynTextBench framework as well as on real-world tasks. Specifically, we simulated 20 synthetic datasets as described in Section 2.4 and obtained one goodness function $s(a)$ for each LLM. We plot these functions together in Figure 3, from which the final SynTextBench score can be determined by definition. We refer readers to Appendix Table 6 for the exact numbers due to the page limit. To gauge the performance of these pretrained LLMs on downstream real-world tasks, we evaluate the given models on SentEval (the Evaluation Toolkit for Universal Sentence Representations (Conneau & Kiela, 2018)) and show the detailed numbers in Appendix Table 7 and Figure 7. SentEval tasks include seven semantic textual similarity tasks (denoted by "STS tasks"), where results are given by the Spearman's correlation with output range $[-1, 1]$, and seven transfer learning tasks (denoted by "Transfer task"), where results are given by the standard accuracy with range $[0, 1]$. We scale the former to the same range as the latter, $[0, 1]$, and take an average as the final accuracy indicator.

**Correlation with real-world tasks.** To demonstrate the informativeness of SynTextBench score, we list the Pearson correlation coefficients between real-data-free evaluation methods and the accuracy of SentEval tasks in Table 1. Five real-data-free metrics are considered that includes Val loss, MDL, SDL, $\varepsilon$SC, and the proposed SynTextBench. Since the smaller the baseline metrics are, the better, we add a negative sign in front of them when calculating the Pearson correlation coefficient. As we have the flexibility of generating synthetic datasets with various sizes (number of sentences), we compare four configurations $n = \{4096, 8192, 16384, 32768\}$. From Table 1, we observe that SynTextBench consistently gives scores highly correlated with real-world task accuracy, with correlation coefficients that are above 0.9. For the four baselines, the highest correlation ever achieved is when $n = 8192$ and evaluated by Val loss, 0.65. It is noteworthy that SynTextBench is also a stabler metric as substantiated by the smaller standard deviation.

**Ablation on the nesting structure.** To showcase the effect of the nesting structure, we see that no nesting structure is a special case of our proposed framework when $p_n = 0$ (cf. Equation 1). In Table 1, we have SynTextBench($p_n = 0.5$) = 0.97. In comparison, we run the analysis for $p_n = 0$ and obtain SynTextBench($p_n = 0$) = 0.92. In conclusion, SynTextBench, with both parameters, outperform the baselines by large margins. Between the two, SynTextBench with the imposed structure further improves the correlation.

**Robustness implications.** To understand how real-data-free evaluation methods correlate with real-world task robustness-accuracy performance, we further analyze the correlation with the robustness indicator, the average percentage of perturbed words, on Transfer tasks when $n = 8192$. We focus on these tasks as they are classification tasks where adversarial attacks are well-defined. To combine robustness correlation with accuracy correlation, we add up two ranking vectors by robustness and accuracy measures, and calculate its Pearson correlation with the ranking by one of the real-data-free evaluation metrics (Val loss, MDL, SDL, $\epsilon$SC, SynTextBench). This way, we effectively obtain the aggregated Spearman correlation coefficient between real-data-free evaluation metrics and joint robustness-accuracy performance. We refer readers to Appendix A.9 for more experimental details. We list the results in Table 2. From the "Rob.-Acc." column, we see SynTextBench has an overall higher correlation with robustness-accuracy performance compared to other baselines. To be more precise, SynTextBench shows a coefficient of 0.76, whereas MDL and SDL are 0.64 and 0.60. Recall that accuracy results were aggregated from STS tasks and Transfer tasks. In Table 2, we also show how each component contributes to the correlation. In the "Rob.-STS" and "Rob.-Transfer" columns, we use only STS or Transfer task results as the accuracy measure when ranking the models, and the remaining steps follow. From the two columns, we see that SynTextBench still shows a stronger

correlation compared to baselines, while having a slightly better correlation with Robustness-STS accuracy performance than Robustness-Transfer accuracy performance.

**Case study on model comparisons using SynTextBench.** Besides having high correlation with real-world task performance, we show how SynTextBench can be used to make model comparisons. In Table 3, we list the SynTextBench scores of pretrained T5 and ST5 under different dataset sizes $n$, together with the accuracy and robustness on SentEval tasks. From the table, it can be seen that the SynTextBench score of ST5 is significantly higher than that of T5 across all $n$, indicating contrastive fine-tuning is beneficial for improving sentence embeddings. This conclusion is in sync with the observations from real-world tasks, where we see ST5 yields both higher accuracy and robustness.

Table 3: Performance evaluation of T5 and ST5 by real-data-free metric (SynTextBench) and real-data-dependent metrics (accuracy and robustness on SentEval).

| | SynTextBench | | | | real-world | |
|---|---|---|---|---|---|---|
| n | 4096 | 8192 | 16384 | 32768 | accuracy | robustness |
| T5 | 0.111±0.002 | 0.130±0.001 | 0.145±0.000 | 0.158±0.001 | 82.78 | 12.21 |
| ST5 | 0.214±0.000 | 0.223±0.001 | 0.227±0.001 | 0.230±0.000 | 90.17 | 13.23 |

### 3.3 Extended study on large language models and in-context learning

We emphasize that SynTextBench focused on analyzing the sentence embeddings of language models, for which larger decoder models generally do not have better performance than smaller encoder models (Ethayarajh, 2019). Therefore, we followed Gao et al. (2021) and mostly conducted sentence embedding experiments with models therein. To demonstrate the general applicability of the framework to various LLM types, we include in this experiment more large decoder language models such as LLaMA and OPT models (Touvron et al., 2023a;b; Zhang et al., 2022). In Appendix Table 8, we calculated the Pearson correlation coefficients between different evaluation methods and the accuracy of SentEval tasks in the right-most column. One sees that, similar to the observations with encoder models, SynTextBench also gives scores highly correlated with real-world task accuracy on decoder models, with the correlation coefficient of 0.871.

Besides probing tasks, we also evaluate the few-shot in-context learning (ICL) performance on SentEval transfer tasks and SynTextBench synthetic task. We do not include STS tasks since they are typically measured by cosine distance, whose ICL prompts are less obvious to us. We also excluded TREC as we have not found proper prompts that could lead to reasonable accuracy. The instructions we give include two demonstrations with one demonstration for each class. For example, in CR (customer review), we use the instruction: "Answer the sentiment of the following review, either Positive or Negative. \n\nQ: We tried it out

Table 4: Correlation between real-data-free evaluation metric and real-data accuracy under ICL settings.

| Name | Pearson correlation |
|---|---|
| Val loss | 0.17 |
| MDL | 0.20 |
| SDL, $\varepsilon = 1$ | 0.15 |
| $\varepsilon$SC, $\varepsilon = 1$ | - |
| SynTextBench-ICL | 0.81 |

christmas night and it worked great .\nA: Positive\n\nQ: very bad quality .\nA: Negative\n\n". We give the accuracy in Appendix Table 9 and the Pearson correlation between the average ICL accuracy and different metrics in Table 4. In the bottom row of Table 4, we calculate the correlation between the ICL accuracy on SynTextBench synthetic task (denoted by SynTextBench-ICL) and the average ICL accuracy on SentEval tasks. We can see that SynTextBench-ICL still shows strong correlation (above 0.8) with ICL accuracy on SentEval tasks, whereas the best baseline only correlates with ICL accuracy with correlation coefficient $\leq 0.2$.

## 4 Conclusion

In this paper, we have proposed SynTextBench, a novel framework for evaluating the accuracy and robustness of LLM sentence embeddings. SynTextBench is a configurable real-date-free lightweight testbed that generates steerable synthetic language datasets and proxy tasks, avoiding the risk of test-data leakage. SynTextBench is the pioneering effort in developing synthetic benchmarking methodologies for NLP, with a primary focus on sentence classification tasks and does not cover other NLP tasks such as question answering, machine translation, or summarization. By concentrating on this specific aspect, we have provided a solid foundation upon which future research can build. We believe that our work is a major step towards ensuring independent and sustainable auditing of LLMs.

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

# A  APPENDIX

## A.1  RELATED WORK AND BACKGROUND

**Sentence representations.** To obtain performant LLMs, learning universal sentence representations that capture rich information for various downstream NLP tasks without task-specific finetuning is an active research field and has also been studied extensively in the past years (Kiros et al., 2015; Conneau et al., 2017; Gao et al., 2019; Li et al., 2020; Su et al., 2021; Giorgi et al., 2021; Gao et al., 2021; Chuang et al., 2022). While learning to extract ideal sentence embeddings, (Gao et al., 2019; Li et al., 2020; Ethayarajh, 2019) have pinpointed the anisotropic behavior in the sentence embedding vector space as a reason behind sentence embeddings' poor capture of semantic information. To remedy the situation, Bert-flow (Li et al., 2020) and Bert-whitening (Su et al., 2021) transformed the sentence embedding distribution into an isotropic Gaussian distribution through normalizing flow and whitening post-processing. Through contrastive learning, SimCSE (Gao et al., 2021) and DiffCSE (Chuang et al., 2022) also achieved new state-of-the-art sentence embedding performance by promoting uniformity and alignment (Wang & Isola, 2020).

**Evaluations of pretrained models.** In evaluating the performance of LLMs, the current de facto evaluation paradigm is to utilize widely-used NLP benchmarks such as the General Language Understanding Evaluation (GLUE (Wang et al., 2018)/SuperGLUE (Wang et al., 2019)) benchmark, the Stanford Question Answering Dataset (SQuAD v1.1 (Rajpurkar et al., 2016)/v2.0 (Rajpurkar et al., 2018)), the Situations With Adversarial Generations (SWAG (Zellers et al., 2018)) dataset, the ReAding Comprehension from Examinations (RACE (Lai et al., 2017)) dataset, the Evaluation Toolkit for Universal Sentence Representations (SentEval (Conneau & Kiela, 2018)), BIG-Bench (Srivastava et al., 2022), etc. In many cases, these NLP benchmarks are supersets of datasets, e.g., GLUE is a collection of 9 datasets for evaluating natural language understanding systems, and SentEval is a collection of 7 Semantic Textual Similarity (STS) tasks and 7 transfer datasets that have partial overlap with GLUE. The heavy reliance on real-world tasks can be exemplified by broad literature. For example, Bert (Devlin et al., 2019) was evaluated on GLUE, SQuAD v1.1/2.0, SWAG; Roberta (Liu et al., 2019) was evaluated on GLUE, SQuAD v1.1/2.0, RACE; and T5 (Raffel et al., 2020) was evaluated on GLUE/SuperGLUE, SQuAD, CNN/Daily Mail abstractive summarization and WMT translation. HELM (Liang et al., 2022) proposes a holistic evaluation framework for language models that measures 7 metrics on 42 scenarios. However, when confronting the challenge of test-data leakage, to the best of our knowledge, there is no real-data-free evaluation method for NLP pretrained representations. In a recent literature (Ko et al., 2022), authors reported the validation loss (Val loss), minimum description length (MDL) (Blier & Ollivier, 2018; Voita & Titov, 2020), surplus description length (SDL) and $\epsilon$-sample complexity ($\epsilon$SC) (Whitney et al., 2020) on class-conditional Gaussian distribution data as an effort to build task-agnostic evaluation baselines for pretrained representations in computer vision. Our proposed framework differs from this line of work in that we focus on the domain of natural language processing and we do not assume the data inputs are sampled from an idealized distribution. Instead, we create synthetic sentences and proxy tasks based on a lexical resource for LLM evaluation.

**Sentiment lexicons.** SentiWordNet 3.0 (Baccianella et al., 2010) is a lexical resource that provides sentiment information for each word in WordNet (Miller, 1995), a widely-used lexical database of English words and their relationships. SentiWordNet 3.0 is an improved version of SentiWordNet 1.0 (Esuli & Sebastiani, 2006), 1.1 (Esuli & Sebastiani, 2007), 2.0 (Esuli, 2008). SentiWordNet automatically assigns synsets of WordNet according to notions of "positivity", "negativity", and "neutrality". The sentiment scores of a synset are assigned on a scale from 0.0 to 1.0 and sum to 1, reflecting a fine-grained opinion-related word-level labeling. SentiWordNet has been used in a variety of natural language processing tasks, such as sentiment analysis (Denecke, 2008; Ohana & Tierney, 2009; Khan et al., 2016), opinion mining (Husnain et al., 2021; Dadhich & Thankachan, 2021), representation learning (Ke et al., 2020), and curriculum learning (Rao et al., 2020). Besides Senti-WrodNet, other sentiment lexicons include Affective Norms for English Words (ANEW) (Bradley & Lang), Warriner lexicon (Warriner et al., 2013), a new ANEW (Nielsen, 2011), and ANEW+ (Shaikh et al., 2016). In this paper, we will demonstrate the use of sentiment lexicon with word-level labels in constructing synthetic datasets using SentiWordNet 3.0; however, the framework proposed in this paper can take any lexicon with word-level labels. We also envision our framework to benefit from a richer vocabulary and extend to other value lexicons like moral lexicons (Rezapour et al., 2019).

**Robust Bayes optimal classifier.** Despite the difficulty of characterizing the optimal classifier with the minimum loss for generic data, for data drawn from class-conditional Gaussian distribution, the explicit optimal strategy is given by Fisher's linear discriminant rule (Johnson et al., 2002; Petridis & Perantonis, 2004). Likewise, the optimal classification strategy can also be given for such data in the presence of input perturbations (Bhagoji et al., 2019; Dan et al., 2020). Let $\mathcal{N}(\mu, \Sigma)$ denote Gaussian distribution with mean $\mu$ and variance $\Sigma$. Generally, for binary classification problems with data pair $(x, y)$ generated from a probability distribution $P_{\mu, \Sigma}$: $x|y = 1 \sim \mathcal{N}(\mu, \Sigma)$, $x|y = -1 \sim \mathcal{N}(-\mu, \Sigma)$, the classifier that minimizes the adversarial loss (Awasthi et al., 2021) $\max_{x': \|x' - x\| \leq \epsilon} \mathbb{1}(f(x') \neq y)$, the robust Bayes optimal classifier (Bhagoji et al., 2019; Dan et al., 2020), is given by $\text{sign}(w_0^T x)$, where $w_0 = \Sigma^{-1}(\mu - z_\Sigma(\mu))$ and $z_\Sigma$ is the solution of the convex problem

$$\arg\min_{\|z\|_2 \leq \epsilon} (\mu - z)^T \Sigma^{-1} (\mu - z) \tag{2}$$

In the following sections, we will exploit robust Bayes optimal classifier in giving the explicit optimal classifier on whitened sentence embeddings and develop our theoretical groundings on top of it.

## A.2 LIST OF STOP WORDS

{'must', 'meanwhile', 'among', 'same', 'you', 'formerly', 'already', 'take', 'he', 'thereupon', 'done', 'anyhow', 'almost', 'ca', 'regarding', 'will', 'mostly', 'say', 'again', 'forty', 'seemed', 'still', 'they', "re', 'seem', 'latter', 'why', 'hers', 'thereby', 'themselves', 'your', 'nine', 'become', 'may', 'beyond', 'it', 'back', 'our', 'himself', "m', 'via', 'we', 'seems', 'throughout', 'yourself', 'bottom', 'only', 'whereby', 'move', 'else', 'front', 'within', 'after', 'every', 'quite', 'hereby', 'now', 'since', 'became', 'herself', 'behind', 'any', 'those', 'used', 'indeed', "ve', 'first', 'moreover', 'ourselves', 'she', 'should', 'her', 'various', 'few', 'hundred', 'whoever', 'give', 'latterly', 'between', 'in', 'most', 'make', 'sixty', 'therefore', "'s", 'hence', 'amount', 'otherwise', "m', "re', "s', 'are', 'could', 'along', 'ours', 'of', 'that', 'everywhere', 'during', 'his', 'then', 'fifty', 'namely', 'when', 'around', 'all', 'keep', 'these', "ll', 'third', 'being', 'thus', 'more', "s', 'is', 'where', 'further', 'them', 'towards', 'next', 'and', 'a', 'does', 'here', 'ten', 'whom', 'except', 'myself', 'somehow', 'ever', 'enough', 'there', 'mine', 'other', 'so', 'hereupon', 'who', 'eight', 'one', 'hereafter', 'amongst', 'seeming', 'its', 'each', 'sometime', 'this', 'me', "ll', 'until', 'him', 'because', 'many', 'anyway', 'part', 'from', 'have', 'over', 'to', "'re', 'becomes', 'too', 'as', 'name', 'whence', 'whole', 'herein', 'everything', 'against', 'call', 'upon', 'both', 'i', 'whenever', 'across', 'anywhere', 'six', 'us', 'thereafter', 'also', 'former', 'whither', 'whose', 'such', 'really', 'was', "d', 'someone', "ve', 'eleven', 'wherein', 'yours', 'by', 'their', 'beside', 'or', 're', 'has', 'off', 'which', 'put', 'whether', 'per', 'four', 'whereafter', 'often', 'doing', 'had', 'out', 'some', 'fifteen', 'others', 'once', 'somewhere', 'either', 'besides', 'though', 'been', 'do', 'very', 'thru', 'go', 'please', 'sometimes', "'ll', 'perhaps', 'whereupon', 'whatever', 'about', 'for', 'itself', 'thence', 'at', 'how', 'made', 'three', 'might', 'another', 'did', 'alone', 'elsewhere', 'toward', 'were', 'would', 'due', 'what', 'an', 'wherever', 'be', 'can', 'something', 'side', "'d', 'with', "'m", 'am', 'therein', 'into', 'through', "'ve', 'everyone', 'on', 'my', 'even', 'own', 'see', 'several', 'two', 'afterwards', 'show', "d', 'beforehand', 'nowhere', 'becoming', 'last', 'onto', 'the', 'yourselves', 'five', 'anyone', 'together', 'before', 'always', 'get', 'using'}

## A.3 SENTIWORDNET 3.0 SYNSETS

We drop columns POS, ID, GLOSS in the examples for easier illustration. By performing the procedure on synsets in Table 5, we obtain a positive word list {able, living, accurate, concrete, active}, a negative word list {unfaithful, unable}, a neutral word list {acroscopic, straight}.

Table 5: Examples of synsets in SentiWordNet 3.0.

| SynsetTerms | PosScore | NegScore | SynsetTerms | PosScore | NegScore |
|---|---|---|---|---|---|
| able#1 | 0.125 | 0 | unable#1 | 0 | 0.75 |
| acroscopic#1 | 0 | 0 | unquestioning#2 | 0.5 | 0.5 |
| living#3 | 0.5 | 0.125 | concrete#1 | 0.625 | 0.25 |
| accurate#1 | 0.5 | 0 | straight#5 | 0 | 0 |
| unfaithful#4 | 0 | 0.5 | active#5 | 0.5 | 0.125 |

## A.4 SYNTHETIC SENTENCE EXAMPLES AND DISCUSSIONS

**POSITIVE**

- "perfectibility lotus-eater shine shine health_care health_care pleasant-tasting"
- "convincingly gruesomely gruesomely convincingly deserve feeder exhaust exhaust debonaire stuffily stuffily anne_sexton wholeness wholeness rarefy conformable pretension pretension"
- "smarmily smarmily fairness covetously infuse soothing subtly subtly soothing"
- "precious grace the_right_way the_right_way absoluteness absoluteness"
- "personal_relation pleasurable sleekness cryptographically cryptographically correct delineate sink_in authenticated"
- "perfectibility lotus-eater shine shine health_care health_care pleasant-tasting"

**NEGATIVE**

- "unpleasant unpleasant mortal sympathetic dead dead choker nubbly fallout"
- "counterrevolutionary apprehensive thunderclap unskilled unskilled thunderclap apprehensive cheat shanny shanny cheat counterrevolutionary smooth smooth decayed decayed imagine imagine loser unpicturesque unnaturalized unnaturalized unrelieved unrelieved unhewn"
- "unpleasant unpleasant mortal sympathetic dead dead choker nubbly fallout"
- 'jostling weka offend engorged fouled fouled engorged intermittence space impaction impaction space intermittence dishonesty disgustingly"
- "blindly blindly"
- "second_class criminal_possession lousiness nonextensile linanthus_dianthiflorus nonarbitrary regular foolishness stabbing"

**Discussions on non-grammatical sentences.** As we mentioned earlier in the paper, the inclusion of the parenthesis is to guarantee we test for the linguistic structures, whose importance is repeatedly advocated in literatures from both machine learning and cognitive science. Therefore, when building synthetic test for the linguistic structures, we also follow the parenthesis and thus have non-grammatical test sets.

We would like to motivate their use based on the following example of sentiment analysis in food reviews. Upon seeing the review "love love fantastic!" in a food review, a reasonable language model should recognize the entailed positive sentiment, even though the sentence is non-grammatical. In our framework, to test the other basic skill for language acquisition in a systematic and scalable manner, we put words associated with binary labels (positive and negative) in the synthetic sentence and test sentence embeddings of LLMs in identifying the words for sentence classification. Related to our setups herein, Krishna et al. (2021) also studies a range of summarization tasks from nonsense documents, in which a task is also designed to classify whether there are keywords indicating positive or negative sentiments (Krishna et al. (2021), Figure 1). Additional evidence of the usage of non-grammatical sentences can be found in Bhatia et al. (2023), where authors also exploit non-grammartical synthetic sentence (Bhatia et al. (2023), Appendix A) for constructing Gaussian logistic regression problems in improving reasoning ability in LMs, which manifests the value of non-grammatical language in learning/testing basic skills. Our high correlation with real-world tasks further suggests that better understanding of the synthetic sentences indeed implies better performance on real tasks. By construction, our framework is not limited to sentiment analysis as one can readily change the base lexicon to test how LLMs identify words describing other notions. For example, if we use the moral foundation lexicon, one can test how each LLM identifies words that describe care, fairness, loyalty, authority, and sanctity.

A.5   HISTOGRAMS OF SYNTHETIC DATASETS VERSUS ENGLISH WIKIPEDIA CORPUS

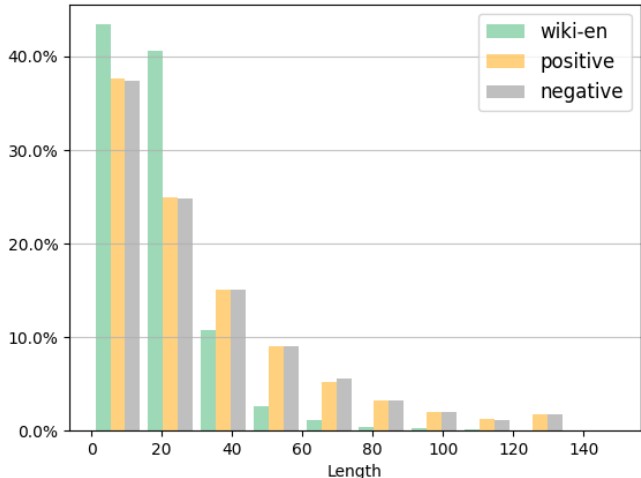

Figure 4: The histograms of sentence lengths in the English Wikipedia corpus (stop words removed) and the constructed synthetic corpus (positive/negative sentences).

A.6   TASK FEASIBILITY

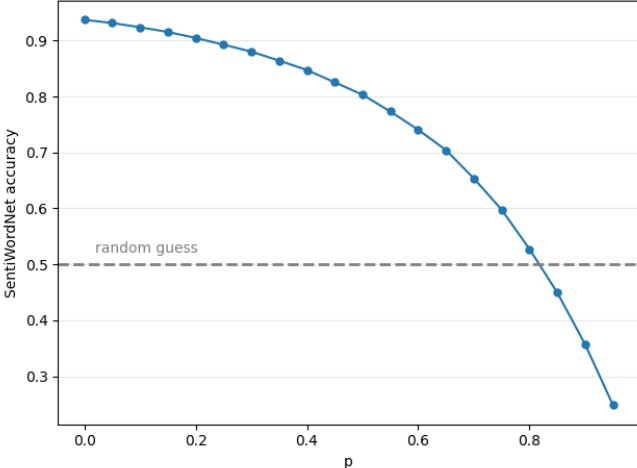

Figure 5: The reference accuracy given by SentiWordNet sentiment analysis. With an increasing mixing ratio $p$, the task becomes harder and the reference accuracy also shows a decreasing trend.

A.7 ROBUST BAYES OPTIMAL CLASSIFIER AND PROOFS

To motivate our findings, we first plot the Bayes optimal robust classifiers together with the Bayes optimal classifier in three 2D cases in Figure 6. From the plot, we see that as long as the direction of $\mu$ is in parallel to one of the two eigenvectors, the robust Bayes optimal classifiers would overlap with the Bayes optimal classifier.

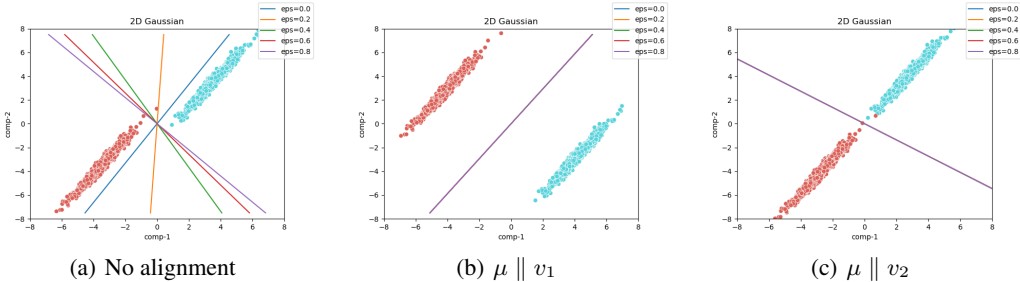

(a) No alignment        (b) $\mu \parallel v_1$        (c) $\mu \parallel v_2$

Figure 6: Three 2D examples of the Bayes optimal classifier and robust Bayes optimal classifiers with different magnitudes of expected perturbation $\epsilon$. Figure 6(a) - no alignment between the mean vector $\mu$ and the eigenvectors. Figure 6(b) and Figure 6(c) - $\mu$ is parallel to the eigenvector corresponding to either of the two eigenvalues.

To generalize the result, we prove the following theorem that specifies a sufficient condition for all $\epsilon$-robust Bayes optimal classifiers to overlap with each other (including $\epsilon = 0$, i.e. Bayes optimal classifier). Intuitively, if the $\epsilon$-robust Bayes optimal classifiers overlap with the Bayes optimal classifiers, then there is no robustness-accuracy trade-off.

**Result A.1.** The $\epsilon$-robust Bayes optimal classifiers overlap for all $\epsilon$ if the vector difference $\mu$ between the centers of the two gaussians lies completely within a degenerate subspace of the eigenspace of the covariance matrix, i.e. with eigenpairs $\{(\lambda_k, v_k), k \in [n]\}$, for $\forall i, j \in \{k : \lambda_k \neq 0, \mu^T v_k \neq 0\}$, $\lambda_i = \lambda_j = \lambda$.

*Proof.* Let $v_1, \ldots, v_n$ and $\lambda_1, \ldots, \lambda_n$ be the orthonormal eigenbasis and the corresponding eigenvalues of the covariance matrix $\Sigma$, then we have $\Sigma^{-1} = \sum_{i=1}^{n} \frac{1}{\lambda_i} v_i v_i^T$. Following Dan et al. (2020), we see that the $\epsilon$-robust classifier is given as $\text{sign } w^{\epsilon \top} x$, where $w^\epsilon = \Sigma^{-1}(\mu - z_\Sigma^\epsilon(\mu))$ and

$$z_\Sigma^\epsilon(\mu) = \underset{\|z\| \leq \epsilon}{\arg\min} \|\mu - z\|_{\Sigma^{-1}}^2.$$

Let $\mu = \sum_{i=1}^{n} a_i v_i$ and we re-parameterize $z = \sum_{i=1}^{n} b_i v_i$. Then,

$$z_\Sigma^\epsilon(\mu) = \sum_{i=1}^{n} b_i^\epsilon v_i, \quad \text{where } b^\epsilon = \langle b_i^\epsilon \rangle_{i=1}^n = \underset{\sum_{i=1}^{n} b_i^2 \leq \epsilon^2}{\arg\min} \sum_{i=1}^{n} \frac{(a_i - b_i)^2}{\lambda_i}$$

By using the Lagrange multiplier $\gamma_\epsilon$ with first-order optimality condition, we see that $\forall i$

$$\frac{b_i^\epsilon - a_i}{\lambda_i} + \gamma_\epsilon b_i^\epsilon = 0 \iff \frac{a_i - b_i^\epsilon}{\lambda_i} = \gamma_\epsilon b_i^\epsilon \iff b_i^\epsilon = \frac{a_i}{1 + \lambda_i \gamma_\epsilon} \quad (3)$$

and $\sum_{i=1}^{n} (b_i^\epsilon)^2 \leq \epsilon^2$. In order for all the robust classifiers to overlap we need $w^\epsilon / \|w^\epsilon\|$ to the independent of $\epsilon$. That is,

$$\frac{w^\epsilon}{\|w^\epsilon\|} = \frac{\sum_{i=1}^{n} v_i \frac{a_i - b_i^\epsilon}{\lambda_i}}{\sqrt{\sum_{i=1}^{n} \left(\frac{a_i - b_i^\epsilon}{\lambda_i}\right)^2}} = \frac{\sum_{i=1}^{n} \gamma^\epsilon b_i^\epsilon v_i}{\sqrt{\sum_{i=1}^{n} (\gamma^\epsilon)^2 (b_i^\epsilon)^2}} = \frac{\sum_{i=1}^{n} b_i^\epsilon v_i}{\sqrt{\sum_{i=1}^{n} (b_i^\epsilon)^2}} = \frac{\sum_{i \in S} b_i^\epsilon v_i}{\sqrt{\sum_{i \in S} (b_i^\epsilon)^2}},$$

where the $S$ in the last equation denotes the set of indices for which $a_i \neq 0$. For $\forall i$ with $a_i = 0$, from equation 3, we clearly have $b_i^\epsilon = 0$.

The condition $\mu$ lies completely within a degenerate subspace of the eigenspace of $\Sigma$ is equivalent to saying $\lambda_i = \lambda_j = \lambda$ for $\forall\, i, j \in S$. In this case, we see that for $\forall\, i \in S$,

$$\epsilon^2 \geq \sum_{i=1}^{n} (b_i^{\epsilon})^2 = \sum_{i \in S} (b_i^{\epsilon})^2 = \left(\frac{1}{1 + \lambda\gamma_\epsilon}\right)^2 \sum_{i \in S} a_i^2,$$

so $\frac{1}{1+\lambda\gamma_\epsilon} \leq \epsilon \frac{1}{\sqrt{\sum_{i \in S} a_i^2}}$, $b_i^{\epsilon} \leq \frac{\epsilon}{\sqrt{\sum_{i \in S} a_i^2}} a_i$. So, we get $b_i^{\epsilon} = m_\epsilon \cdot a_i$ where $m_\epsilon = \min\left(1, \frac{\epsilon}{\sqrt{\sum_{i \in S} a_i^2}}\right)$

$$\frac{w^\epsilon}{\|w^\epsilon\|} = \frac{\sum_{i \in S} b_i^\epsilon v_i}{\sqrt{\sum_{i \in S}^{n} (b_i^\epsilon)^2}} = \frac{\sum_{i \in S} m_\epsilon a_i v_i}{m_\epsilon \sqrt{\sum_{i \in S} a_i^2}} = \sum_{i \in S} \frac{a_i}{\sqrt{\sum_{i \in S} (a_i)^2}} v_i,$$

which is independent of $\epsilon$. $\qquad\square$

**_Result_ A.2.** Consider the robust Bayes optimal classifier[1], $f_\epsilon$, for $P_{\mu_1, \mu_2, I_d}$ with class prior $\mathbb{P}(y = 1) = \tau$, $\mathbb{P}(y = -1) = 1 - \tau$, it is in the following form

$$f_\epsilon(x) = \text{sign}\left\{\left(x - \frac{\mu_1 + \mu_2}{2}\right)^T \tilde{\mu}(1 - \epsilon/\|\tilde{\mu}\|_2) - q/2\right\},$$

where $\tilde{\mu} = \frac{\mu_1 - \mu_2}{2}$ and $q = ln\{(1 - \tau)/\tau\}$. For any sample $x$, $f_\epsilon$ gives the lower bound on the decision margin $\delta$

$$\left(x + \delta - \frac{\mu_1 + \mu_2}{2}\right)^T \tilde{\mu}(1 - \epsilon/\|\tilde{\mu}\|_2) - q/2 = 0$$

$$\Leftrightarrow \quad \delta^T \tilde{\mu}(1 - \epsilon/\|\tilde{\mu}\|_2) = q/2 - \left(x - \frac{\mu_1 + \mu_2}{2}\right)^T \tilde{\mu}(1 - \epsilon/\|\tilde{\mu}\|_2)$$

$$\Rightarrow \quad \|\delta\|_2 \geq \frac{|(x - \frac{\mu_1 + \mu_2}{2})^T \tilde{\mu}(1 - \epsilon/\|\tilde{\mu}\|_2) - q/2|}{\|\tilde{\mu}(1 - \epsilon/\|\tilde{\mu}\|_2)\|_2},$$

which then yields the worst-case bound

$$\|\Delta\|_2 = \min\|\delta\|_2 = \frac{|(x - \frac{\mu_1 + \mu_2}{2})^T \tilde{\mu}(1 - \epsilon/\|\tilde{\mu}\|_2) - q/2|}{\|\tilde{\mu}(1 - \epsilon/\|\tilde{\mu}\|_2)\|_2}.$$

Since the bound $\|\Delta\|_2$ is subject to the positions of two Gaussians, we scale the bound by the distance from Gaussian centers to the classifier. We note that, since the class are imbalanced, the distances from the two Gaussian centers to the classifier $f_\epsilon$ are different, i.e. $\frac{|\tilde{\mu}^T \tilde{\mu}(1 - \epsilon/\|\tilde{\mu}\|_2) - q/2|}{\|\tilde{\mu}(1 - \epsilon/\|\tilde{\mu}\|_2)\|_2}$ and $\frac{|\tilde{\mu}^T \tilde{\mu}(1 - \epsilon/\|\tilde{\mu}\|_2) + q/2|}{\|\tilde{\mu}(1 - \epsilon/\|\tilde{\mu}\|_2)\|_2}$, respectively. We hereby take their average as the scaling factor and obtain

$$\|\bar{\Delta}\|_2 = \frac{|(x - \frac{\mu_1 + \mu_2}{2})^T \tilde{\mu}(1 - \epsilon/\|\tilde{\mu}\|_2) - q/2|}{\|\tilde{\mu}(1 - \epsilon/\|\tilde{\mu}\|_2)\|_2} \frac{2\|\tilde{\mu}(1 - \epsilon/\|\tilde{\mu}\|_2)\|_2}{|\tilde{\mu}^T \tilde{\mu}(1 - \epsilon/\|\tilde{\mu}\|_2) - q/2| + |\tilde{\mu}^T \tilde{\mu}(1 - \epsilon/\|\tilde{\mu}\|_2) + q/2|}$$

$$= \frac{2|(x - \frac{\mu_1 + \mu_2}{2})^T \tilde{\mu}(1 - \epsilon/\|\tilde{\mu}\|_2) - q/2|}{|\tilde{\mu}^T \tilde{\mu}(1 - \epsilon/\|\tilde{\mu}\|_2) - q/2| + |\tilde{\mu}^T \tilde{\mu}(1 - \epsilon/\|\tilde{\mu}\|_2) + q/2|}.$$

---

[1]Dobriban, E., Hassani, H., Hong, D. and Robey, A., 2020. Provable tradeoffs in adversarially robust classification. arXiv preprint arXiv:2006.05161.

## A.8 COMPLETE RESULTS

Table 6: Pearson correlation comparison between real-data-free evaluation methods and the average accuracy on the real-world tasks included in Table 7. Since the smaller the Val loss, MDL, SDL and $\epsilon$SC, the better, we add a negative sign in front of them when calculating the Pearson correlation coefficient.

| n | Name | BERT$_{base}$ | DiffCSE-B | BERT$_{large}$ | T5$_{base}$ | T5$_{large}$ | RoBERTa$_{base}$ | DiffCSE-R | GPT | ST5 | Pearson |
|---|---|---|---|---|---|---|---|---|---|---|---|
| | Reallife acc. | 83.50 | 86.81 | 83.68 | 82.78 | 82.36 | 83.83 | 88.19 | 78.01 | 90.17 | 1.0 |
| 4096 | Val loss | 1.0e-06±1e-07 | 1.4e-06±3e-07 | 7.6e-07±5e-08 | 8.5e-08±1e-08 | 5.4e-08±9e-09 | 4.0e-06±3e-07 | 1.1e-06±8e-08 | 3.1e-03±8e-04 | 3.7e-03±5e-03 | 0.285±0.498 |
| | MDL | 5002±318 | 4755±129 | 5422±357 | 7318±119 | 6724±228 | 5396±181 | 4773±296 | 5604±366 | 4433±360 | 0.571±0.109 |
| | SDL, $\varepsilon$=1 | 3090±318 | 2843±129 | 3510±357 | 5406±119 | 4812±228 | 3484±181 | 2861±296 | 3687±366 | 2514±368 | 0.570±0.110 |
| | $\varepsilon$SC, $\varepsilon$=1 | 3686±0 | 3686±0 | 3686±0 | 3686±0 | 3686±0 | 3686±0 | 3686±0 | 3686±0 | 3686±0 | - |
| | SynTextBench | 0.137±0.001 | 0.148±0.001 | 0.135±0.000 | 0.111±0.002 | 0.103±0.002 | 0.119±0.001 | 0.193±0.001 | 0.090±0.003 | 0.214±0.000 | 0.939±0.008 |
| 8192 | Val loss | 3.3e-06±3e-07 | 6.3e-04±9e-04 | 6.6e-04±9e-04 | 3.3e-07±9e-08 | 5.9e-04±8e-04 | 1.3e-05±1e-06 | 4.1e-06±2e-07 | 3.1e-02±1e-03 | 1.2e-03±5e-05 | 0.649±0.004 |
| | MDL | 8802±99 | 8687±260 | 10107±156 | 14664±464 | 14487±426 | 9801±489 | 8902±175 | 10001±291 | 7310±175 | 0.519±0.043 |
| | SDL, $\varepsilon$=1 | 5262±99 | 5144±262 | 6564±155 | 11124±464 | 10944±426 | 6261±489 | 5362±175 | 6343±287 | 3766±175 | 0.509±0.043 |
| | $\varepsilon$SC, $\varepsilon$=1 | 7372±0 | 7372±0 | 7372±0 | 7372±0 | 7372±0 | 7372±0 | 7372±0 | 7372±0 | 7372±0 | - |
| | SynTextBench | 0.152±0.001 | 0.156±0.001 | 0.148±0.002 | 0.130±0.001 | 0.122±0.000 | 0.129±0.002 | 0.196±0.001 | 0.085±0.003 | 0.223±0.001 | 0.968±0.006 |
| 16384 | Val loss | 2.3e-03±2e-03 | 9.5e-04±7e-04 | 7.2e-04±1e-03 | 6.6e-04±9e-04 | 1.2e-03±9e-05 | 8.2e-04±1e-03 | 2.2e-03±2e-03 | 2.1e-01±3e-02 | 2.3e-02±9e-04 | 0.605±0.007 |
| | MDL | 15840±436 | 15253±455 | 18039±778 | 26004±879 | 25606±767 | 16629±117 | 15465±349 | 16794±440 | 11895±89 | 0.506±0.032 |
| | SDL, $\varepsilon$=1 | 9266±429 | 8689±458 | 11477±786 | 19443±887 | 19040±767 | 10066±118 | 8891±365 | 8525±383 | 5153±93 | 0.425±0.021 |
| | $\varepsilon$SC, $\varepsilon$=1 | 14745±0 | 14745±0 | 14745±0 | 14745±0 | 14745±0 | 14745±0 | 14745±0 | 14745±0 | 14745±0 | - |
| | SynTextBench | 0.161±0.000 | 0.164±0.001 | 0.161±0.001 | 0.145±0.000 | 0.141±0.001 | 0.137±0.000 | 0.198±0.001 | 0.087±0.001 | 0.227±0.001 | 0.958±0.002 |
| 32768 | Val loss | 6.4e-03±8e-04 | 4.2e-03±2e-03 | 4.1e-03±3e-04 | 3.1e-02±1e-02 | 3.0e-03±7e-04 | 1.4e-02±2e-03 | 1.1e-02±1e-02 | 4.7e-01±2e-02 | 2.9e-01±1e-02 | 0.267±0.018 |
| | MDL | 27667±294 | 25793±898 | 29577±253 | 43955±1616 | 39692±1520 | 27151±33 | 27546±646 | 28930±471 | 21999±88 | 0.481±0.029 |
| | SDL, $\varepsilon$=1 | 15417±282 | 13581±927 | 17367±252 | 31282±1860 | 27501±1518 | 14775±50 | 15214±489 | 9442±195 | 6076±106 | 0.311±0.008 |
| | $\varepsilon$SC, $\varepsilon$=1 | 29491±0 | 29491±0 | 29491±0 | 29491±0 | 29491±0 | 29491±0 | 29491±0 | 12139±0 | 12139±0 | -0.044±0.000 |
| | SynTextBench | 0.170±0.001 | 0.169±0.000 | 0.173±0.001 | 0.158±0.001 | 0.156±0.000 | 0.140±0.001 | 0.202±0.000 | 0.092±0.001 | 0.230±0.000 | 0.934±0.002 |

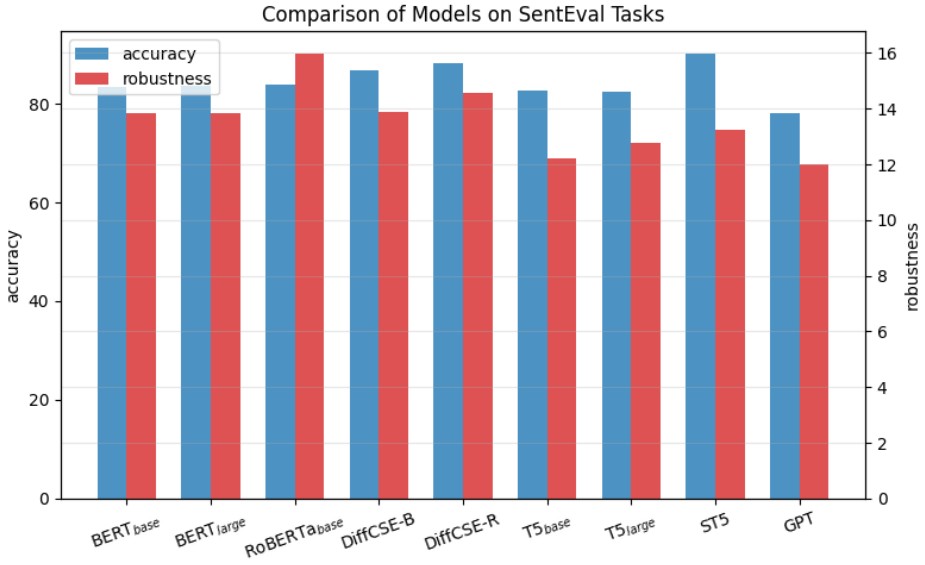

Figure 7: The accuracy and robustness (average percentage of perturbed words) performance of pretrained models on SentEval tasks.

Table 7: The detailed SentEval task performance. For STS tasks, we report Spearman's correlation (%), and for Transfer task, we report the standard accuracy (%).

| | STS tasks | | | | | | | Transfer tasks | | | | | | | |
|---|---|---|---|---|---|---|---|---|---|---|---|---|---|---|---|
| Models | STS12 | STS13 | STS14 | STS15 | STS16 | STS-B | SICK-R | MR | CR | SUBJ | MPQA | SST | TREC | MRPC | avg. |
| SynTextBench corr. | 0.878 | 0.864 | 0.869 | 0.840 | 0.936 | 0.897 | 0.952 | 0.688 | 0.792 | 0.419 | 0.648 | 0.517 | 0.340 | 0.626 | |
| BERT$_{base}$ | 54.44 | 58.03 | 58.86 | 67.94 | 68.42 | 53.88 | 62.06 | 82.98 | 89.56 | 95.43 | 89.92 | 85.45 | 89.8 | 74.03 | 83.50 |
| DiffCSE-B | 68.88 | 76.21 | 73.88 | 79.76 | 78.84 | 75.51 | 67.70 | 82.2 | 88.11 | 95.44 | 91.03 | 84.46 | 88 | 75.71 | 86.81 |
| BERT$_{large}$ | 53.33 | 56.86 | 56.23 | 63.43 | 66.69 | 54.43 | 58.06 | 85.96 | 89.59 | 96.43 | 90.96 | 89.13 | 91.8 | 73.16 | 83.68 |
| T5$_{base}$ | 58.18 | 63.78 | 64.14 | 71.83 | 68.94 | 60.17 | 58.77 | 80.54 | 88.34 | 93.04 | 89.73 | 81.27 | 85.8 | 67.36 | 82.78 |
| T5$_{large}$ | 58.34 | 62.59 | 63.50 | 71.36 | 67.88 | 59.67 | 58.02 | 79.31 | 86.86 | 93.53 | 90.43 | 80.72 | 82.8 | 68.75 | 82.36 |
| RoBERTa$_{base}$ | 57.28 | 55.21 | 59.76 | 69.22 | 64.64 | 58.55 | 61.63 | 84.08 | 86.91 | 95.63 | 89.52 | 88.25 | 91.6 | 74.49 | 83.83 |
| DiffCSE-R | 69.77 | 78.70 | 76.08 | 81.75 | 80.86 | 81.17 | 70.34 | 84.75 | 90.99 | 95.2 | 89.75 | 87.92 | 89.4 | 77.28 | 88.19 |
| GPT | 44.16 | 23.99 | 34.73 | 40.78 | 55.11 | 41.05 | 43.65 | 81.08 | 88.53 | 92.81 | 87.87 | 86.6 | 93 | 70.49 | 78.01 |
| ST5 | 74.32 | 82.83 | 81.50 | 86.14 | 85.95 | 86.04 | 79.76 | 85.88 | 91.81 | 94.4 | 91.09 | 90.88 | 95.8 | 74.26 | 90.17 |

Table 8: The SynTextBench-Score and other baselines of decoder models.

| $n$ | Name | LLaMA-7B | LLaMA-13B | LLaMA-30B | LLaMA-2-7B | LLaMA-2-13B | OPT-13B | OPT-30B | Pearson |
|---|---|---|---|---|---|---|---|---|---|
| | Reallife acc. | 64.55 | 63.78 | 64.53 | 76.13 | 78.51 | 79.72 | 79.44 | 1.0 |
| 8192 | Val loss | 0.036141 | 0.149492 | 0.075583 | 0.000002 | 0.0 | 0.010351 | 0.00362 | -0.803 |
| | MDL | 8114.26 | 7434.78 | 6920.22 | 10331.5 | 9331.91 | 7874.07 | 7589.82 | 0.466 |
| | SDL, $\varepsilon = 1$ | 4435.77 | 3321.93 | 3090.58 | 6791.49 | 5791.91 | 4294.41 | 4035.95 | 0.548 |
| | $\varepsilon$SC, $\varepsilon = 1$ | 7372 | 7372 | 7372 | 7372 | 7372 | 7372 | 7372 | - |
| | SynTextBench | 0.062 | 0.027 | 0.048 | 0.097 | 0.075 | 0.089 | 0.093 | 0.871 |

Table 9: In-context learning accuracy of decoder models.

| | Transfer tasks | | | | | | |
|---|---|---|---|---|---|---|---|
| Models | CR | MR | MPQA | SUBJ | SST2 | MRPC | avg. |
| LLaMA-7B | 85.35 | 90.49 | 74.34 | 48.97 | 88.47 | 53.86 | 73.58 |
| LLaMA-13B | 91.07 | 62.78 | 70.07 | 50.02 | 69.74 | 66.20 | 68.31 |
| LLaMA-30B | 91.97 | 92.60 | 83.77 | 50.01 | 95.83 | 66.26 | 80.07 |
| LLaMA-2-7B | 90.83 | 53.25 | 47.06 | 81.60 | 71.00 | 66.49 | 68.37 |
| LLaMA-2-13B | 91.84 | 91.92 | 80.26 | 52.73 | 95.55 | 66.49 | 79.80 |
| OPT-13B | 90.01 | 69.66 | 69.92 | 49.85 | 76.99 | 66.49 | 70.49 |
| OPT-30B | 90.78 | 82.04 | 63.56 | 50.00 | 87.10 | 66.61 | 73.35 |

## A.9 EXPERIMENTAL DETAILS

When we calculate the correlation between real-data-free evaluation methods and real-world task robustness-accuracy performance, we need to aggregate two metrics - accuracy and robustness. For this purpose, we can obtain a ranking of the models according to the accuracy measure, $R_1$, and a ranking of the models according to the robustness measure, $R_2$. We aggregate two rankings by the simple and commonly-used mean aggregation[2] which yields the overall ranking of models based on accuracy-robustness performance, $R_{\mathrm{ref}}$. On the other hand, we can obtain another ranking of models based on one of the real-data-free evaluation methods (e.g. Val loss, MDL, SDL, $\epsilon$SC, SynTextBench), $R$. Lastly, we calculate the Pearson correlation coefficient between $R$ and $R_{\mathrm{ref}}$.

Moreover, when we calculate the robustness measures, we only perform attacks on Transfer tasks as they are classification tasks where adversarial attacks are well-defined. Since we use the average percentage of perturbed words by PWWS attacks (Ren et al., 2019) as the robustness indicator, we also excluded MPQA and TREC due to their short sentence lengths (MPQA and TREC average sentence lengths are 3.03 and 6.48, respectively). PWWS attacks focus on the text adversarial example generation that could guarantee little semantic shifting and therefore rarely cause ground truth label change (also lexical and grammatical correctness). To meet the semantic constraint, PWWS replaces words in the input texts with synonyms and replace named entities (NEs) with similar NEs to generate adversarial samples. Synonyms for each word can be found in WordNet, a large lexical database for the English language. NE refers to an entity that has a specific meaning in the sample text, such as a person's name, a location, an organization, or a proper noun. Replacement of an NE with a similar NE imposes a slight change in semantics but invokes no lexical or grammatical changes.

We list the robustness results in the following table:

Table 10: The robustness (average percentage of perturbed words) of pretrained representations on Transfer tasks.

| Models | MR | CR | SUBJ | SST | MRPC | avg. |
|---|---|---|---|---|---|---|
| BERT$_{\mathrm{base}}$ | 14.48 | 13.99 | 20.2 | 15.07 | 5.45 | 13.838 |
| DiffCSE-B | 14.46 | 14.7 | 18.64 | 15.19 | 6.39 | 13.876 |
| BERT$_{\mathrm{large}}$ | 14.3 | 14.22 | 19.87 | 15.46 | 5.26 | 13.822 |
| T5$_{\mathrm{base}}$ | 12.71 | 12.82 | 16.8 | 13.66 | 5.05 | 12.208 |
| T5$_{\mathrm{large}}$ | 13.67 | 14.28 | 16.93 | 13.82 | 5.17 | 12.774 |
| RoBERTa$_{\mathrm{base}}$ | 16.4 | 18.35 | 20.74 | 17.26 | 7.12 | 15.974 |
| DiffCSE-R | 15.72 | 16.07 | 18.53 | 16.82 | 5.68 | 14.564 |
| GPT | 12.53 | 13.11 | 15.75 | 13.52 | 5.17 | 12.016 |
| ST5 | 13.6 | 13.08 | 18.36 | 14.22 | 6.9 | 13.232 |

---

[2]Wald, R., Khoshgoftaar, T.M. and Dittman, D., 2012, December. Mean aggregation versus robust rank aggregation for ensemble gene selection. In 2012 11th international conference on machine learning and applications (Vol. 1, pp. 63-69). IEEE.

We also list the ranking of models from different metrics in the following table.

Table 11: Ranking of models from different metrics at $n = 8192$.

| Name | BERT$_{base}$ | DiffCSE-B | BERT$_{large}$ | T5$_{base}$ | T5$_{large}$ | RoBERTa$_{base}$ | DiffCSE-R | GPT | ST5 |
|---|---|---|---|---|---|---|---|---|---|
| Overall accuracy | 6 | 3 | 5 | 7 | 8 | 4 | 2 | 9 | 1 |
| STS accuracy | 7 | 3 | 8 | 4 | 5 | 6 | 2 | 9 | 1 |
| Transfer accuracy | 5 | 6 | 2 | 8 | 9 | 4 | 3 | 7 | 1 |
| Robustness | 4 | 3 | 5 | 8 | 7 | 1 | 2 | 9 | 6 |
| Val loss | 8 | 4 | 3 | 9 | 5 | 6 | 7 | 1 | 2 |
| MDL | 7 | 8 | 3 | 1 | 2 | 5 | 6 | 4 | 9 |
| SDL, $\varepsilon$=1 | 7 | 8 | 3 | 1 | 2 | 5 | 6 | 4 | 9 |
| $\varepsilon$SC, $\varepsilon$=1 | 5 | 5 | 5 | 5 | 5 | 5 | 5 | 5 | 5 |
| SynTextBench | 4 | 3 | 5 | 6 | 8 | 7 | 2 | 9 | 1 |

For example, to calculate SynTextBench correlation with robustness-and-accuracy performance, we calculate the Pearson correlation between (row "Overall accuracy" + row "Robustness") / 2 and "SynTextBench". To calculate SynTextBench correlation with robustness-and-STS accuracy performance, we calculate the Pearson correlation between (row "STS accuracy" + row "Robustness") / 2 and "SynTextBench". To calculate SynTextBench correlation with robustness-and-Transfer accuracy performance, we calculate the Pearson correlation between (row "Transfer accuracy" + row "Robustness") / 2 and "SynTextBench". We note that in all our results prior to Table 11, we always infer the correlation in individual runs before we take an average over all trials. Different from that, the rankings from Val loss, MDL, SDL, $\epsilon$SC, and SynTextBench in Table 11, are inferred from the average metric results over 3 trails for an easier illustration. Therefore, the ranking correlation suggested by the table might have some deviation from what is shown in Table 2.

