# OpenReview forum: "On Robustness-Accuracy Characterization of Large Language Models using Synthetic Datasets"
_ICLR.cc/2024/Conference — ICLR 2024 Conference Withdrawn Submission_

### Official Review · Reviewer_QVm9 · 2023-11-01

**Soundness:** 2 fair
**Presentation:** 1 poor
**Contribution:** 2 fair
**Rating:** 3
**Confidence:** 2

**Summary:**

This paper proposes a new strategy to build synthetic datasets which can be used to evaluate how robust and accurate language models are.
The authors first propose 4 desiderata for language modeling benchmarks:
* **Task substance.** Tasks should test whether large language models encode different classes in a way which is linearly separable.
* **Task difficulty.** Task difficulty should be configurable.
* **Task feasibility.** Tasks should be feasible.
* **Task independence.** Tasks’ groundtruths should be independent of the evaluated models.
* **Task equity.** Tasks should be generatable by anyone.
* **Metric informativeness.** Evaluation should give a clear indication of model quality.

The authors then propose a new synthetic dataset generation process which they argue meets them. They generate synthetic datasets by:
* Collecting a lexicon with words labeled as having either positive, negative, or neutral sentiment.
* Selecting either positive or negative sentiment, and sampling a sentence which will either have only positive + neutral words, or negative + neutral. Sentence sampling is done word by word and creates a hierarchical parenthetical structure such that once a word is sampled it is later regenerated to “open and close” a parenthesis. Notably, the authors argue that the task can be made easier or harder by changing the probability of sampling  a neutral vs a positive/negative word at each timestep.

They then extract hidden states from language models on these datasets, and derive Bayes optimal classifiers for these embeddings. Their final metric is computed based on the decision margin of these Bayes optimal classifiers.

**Strengths:**

The paper investigates an interesting question, how to evaluate language models.

The metrics derived from these synthetic datasets have a very strong correlation (of over 0.90) with real task performance. This implies the proposed benchmark could be very useful in diagnosing language models performance.

**Weaknesses:**

I found the paper a bit confusing at points, and could not follow the entire logic behind the proposed dataset construction pipeline, or how it was used in practice. This is the primary reason behind the low score in my review.
* First, the authors propose to sample sentences following a hierarchical parenthetical structure. They motivate this by claiming a similar structure is present in natural languages. The models, however, if I understand correctly are evaluated only in their sentiment classification performance. Importantly, a sentence’s sentiment classification label in this synthetic dataset does not require knowledge of this hierarchical structure. So why should this hierarchical structure have an impact on this benchmarking procedure?
* Second, I am not entirely sure whether the authors believe that language models should be finetuned on the proposed synthetic datasets, or simply probed on them without any finetuning. I think this could be made clearer. If the models are indeed simply probed on these datasets, acknowledging the issues related to diagnostic probing (see, Belinkov, 2022 for a review) could be interesting here.
* Third, the authors use a Bayes optimal classifier to evaluate the model's “robustness” as measured via the classifier’s decision margin. But I don’t think they motivate why this would be more relevant than simply looking at the accuracy. Maybe spending some time motivating that decision could be useful.
* Fourth, for the baseline probing metrics (e.g., MDL and SDL), were these metrics computed with frozen model parameters (i.e., only training a probe on top of the models parameters)?


Yonatan Belinkov. 2022. Probing Classifiers: Promises, Shortcomings, and Advances. Computational Linguistics, 48(1):207–219.

**Questions:**

See above.

---

### Official Review · Reviewer_ALMM · 2023-11-01

**Soundness:** 3 good
**Presentation:** 3 good
**Contribution:** 2 fair
**Rating:** 3
**Confidence:** 4

**Summary:**

The paper proposed an approach to constructing synthetic data to evaluate LMs, and therefore avoiding test set information leakage when querying LMs via APIs with real-world samples. In particular, the synthetic data is constructed in a rule-based fashion: first generate positive/negative/neutral word lists from SentiWordNet, then sequentially generate sentences with the word lists. The difficulty of the synthetic data is controllable by adjusting the generation process.

**Strengths:**

The motivation is sound. Using synthetic data that can be highly correlated with real-world data to evaluate online LLMs seems like a promising research direction.

**Weaknesses:**

The work has several limitations:

1. *Experiments* The motivation is to avoid/alleviate information leakage when evaluating online LLMs via APIs. However, the experiments are mainly conducted on local LMs such as BERT, RoBERTa, and T5. In addition, the evaluation requires the embedding of LMs, which is not available for most of the current LLM API services.

2. *Tasks* The synthetic task is sentiment analysis and the real-world task (for comparison) used in the paper is SentEval. The task type is monotonous and cannot well distinguish the performance of today's LLMs. For instance, the LLM capabilities of coding and reasoning may not be highly correlated with the evaluation results on the proposed synthetic data.

3. *Robustness* It seems a little bit unnatural to evaluate robustness in the light of the motivation (avoid information leakage).

**Questions:**

N/A

---

### Official Review · Reviewer_XF6K · 2023-11-06

**Soundness:** 4 excellent
**Presentation:** 3 good
**Contribution:** 2 fair
**Rating:** 5
**Confidence:** 3

**Summary:**

1. The paper presents a novel methodology to test the robustness and accuracy of sentence embeddings from large language models
2. The authors first define a desiderata for evaluation metrics for evaluating sentence embeddings from LLMs
3. Using the above desiderata as a guiding principle, the authors propose generating a sentiment classification synthetic task of variable difficulty by leveraging positive, negative and neutral sentiment words. The "sentences" are generated based on a nested paranthesis structure, and the task labels are defined a-priori. The difficulty of a dataset is defined based on the proportion of positive / negative to neutral words in a construction.
4. Based on the dataset, the authors propose evaluating the accuracy and decision margins (as induced by a robust Bayes optimal classifier) for different difficulty levels. This is then further used in a goodness function computing at every an accuracy level (a) the average margin aggregated across the different difficulty levels. The area under the curve (above a certain threshold a_{T}) is then defined to be the metric.
5. The authors demonstrate the utility of the metric, showing strong correlation between the metric and downstream task performance (as measured by the SentEval benchmark) for different different LLMs. The metric also correlates strongly with robustness metrics (as measured by robustness to adversarial attacks). The authors also demonstrate good correlation between in-context learning (ICL) for SentEval tasks and the proposed metric (0.871). Furthermore, the authors also demonstrate good correlation between SynTextBench with ICL and the SentEval ICL (0.81)

**Strengths:**

1. The problem is well motivated and very timely, given the interest in evaluating LLM models.
2. The proposed desiderata serves as a good guiding principle for LLM embedding evaluation. The dataset construction methodology is novel to the best of my knowledge.
3. The proposed SynTextBench metric correlates better than other baseline metrics. The finding that the proposed metric correlates well with task performance on SentEval as well as robustness metrics is also very interesting.

**Weaknesses:**

1. My primary concern is with the limited scope of the paper. The paper primarily considers only evaluating sentence embeddings from LLMs, which while important, is a small part of the overall evaluation landscape of LLMs. Consequently, the title "Robustness-Accuracy characterization of Large Language Models using synthetic datasets" is somewhat misleading. Furthermore, the generation methodology for the synthetic tasks using SentiWordNet for polarity detection does seem somewhat restrictive. For sentence embedding evaluation, it does seem to be a good methodology, but it is not clear how well it would generalize to any generative tasks (e.g. question answering, summarization, etc.). Whether this metric can be leveraged for other tasks (especially for a different class of tasks) needs to be demonstrated in my opinion.

2. While the proposed methodology of using a ratio of positive / negative to neutral sentiment words is a good way of defining difficulty, it does seem somewhat restrictive given the contextual nature of languages. Interesting linguistic phenomena such as sarcasm, irony, etc. are not captured by the proposed methodology, which arguably form for a large part of the difficulty in language understanding especially for such large LLMs. While the authors briefly touch upon the issue of negation, negation in natural language is not limited to structured rules, and any methodology testing the robustness of LLMs should provide a way of capturing this, given that LLMs are generally have a poor understanding of negations ([1]).

3. The baseline metrics are still computed on the synthetic dataset. For a generative LLM model training for example, this potentially results in bad sentence embeddings, which subsequently may result in bad task performance. This is especially problematic when done for a single dataset (as is the case for all the baseline metrics). In contrast, the proposed SynTextBench benefits from aggregating across different difficulty levels, and is somewhat more robust to this issue compared to the baseline metrics. A better way for considering the baselines might be to treat them in the same way as SynTextBench is treated (aggregated across different difficulty levels, thresholded for some value of the metric, and then computing the area under the curve).

4. Additionally, there has been a large amount of work on LLM evaluation [2]. While some of the metrics there do not satisfy the proposed desiderata, it would still be good to see how SynTextBench metric compares to the other metrics proposed in the literature. Concretely, from the paper, it is hard to understand under what conditions should one use SynTextBench over other metrics (eg: say MMLU / Big Bench for language generation).

**Questions:**

1. On page 7, the authors mention they average the output from the first and the last layer as the sentence embeddings. What is the intuition for doing this compared to say just taking either an average of the last layer of the embeddings, or (as done traditionally) using the [CLS] / <S> token embedding ?

2. What is the motivation for using a sentiment polarity task as the base task for constructing the evaluation benchmark ? Concretely, what is the intuition behind why capturing sentiment polarity would result in more utilitarian sentence embeddings (and failing to capture this, especially for non grammatical sentence structures would result in worse performance in real world downstream tasks?)

3. [Minor implementation detail] In Section 3.3, for LLaMa-2-13B models (for example), while computing the SynTextBench metric, is the embedding taken in the same way as that for other decoder models (as described on page 7; the embedding of the last token as sentence embedding)?

---

### Author Response · Authors · 2023-11-19

We thank all the reviewers for their valuable comments. We have decide to withdraw our submission to allow more time for revision.